# Thickness of Biceps and Quadriceps Femoris Muscle Measured Using Point-of-Care Ultrasound as a Representation of Total Skeletal Muscle Mass

**DOI:** 10.3390/jcm11226606

**Published:** 2022-11-08

**Authors:** Rianne N. M. Hogenbirk, Alain R. Viddeleer, Judith E. K. R. Hentzen, Willemijn Y. van der Plas, Cees P. van der Schans, Geertruida H. de Bock, Schelto Kruijff, Joost M. Klaase

**Affiliations:** 1Department of Surgery, University Medical Center Groningen, University of Groningen, 9713 GZ Groningen, The Netherlands; 2Department of Radiology, University Medical Center Groningen, University of Groningen, 9713 GZ Groningen, The Netherlands; 3Department of Surgery, Amsterdam University Medical Center, 1105 AZ Amsterdam, The Netherlands; 4Department of Rehabilitation, University Medical Center Groningen, University of Groningen, 9713 GZ Groningen, The Netherlands; 5Research Group Healthy Ageing, Allied Health Care and Nursing, Center of Expertise Healthy Ageing, Hanze University of Applied Sciences, 9747 AS Groningen, The Netherlands; 6Department of Epidemiology, University Medical Center Groningen, University of Groningen, 9713 GZ Groningen, The Netherlands

**Keywords:** sarcopenia, muscle status, skeletal muscle index, muscle force, POCUS

## Abstract

Generalized loss of muscle mass is associated with increased morbidity and mortality in patients with cancer. The gold standard to measure muscle mass is by using computed tomography (CT). However, the aim of this prospective observational cohort study was to determine whether point-of-care ultrasound (POCUS) could be an easy-to-use, bedside measurement alternative to evaluate muscle status. Patients scheduled for major abdominal cancer surgery with a recent preoperative CT scan available were included. POCUS was used to measure the muscle thickness of mm. biceps brachii, mm. recti femoris, and mm. vasti intermedius 1 day prior to surgery. The total skeletal muscle index (SMI) was derived from patients’ abdominal CT scan at the third lumbar level. Muscle force of the upper and lower extremities was measured using a handheld dynamometer. A total of 165 patients were included (55% male; 65 ± 12 years). All POCUS measurements of muscle thickness had a statistically significant correlation with CT-derived SMI (*r* ≥ 0.48; *p* < 0.001). The strongest correlation between POCUS muscle measurements and SMI was observed when all POCUS muscle groups were added together (*r* = 0.73; *p* < 0.001). Muscle strength had a stronger correlation with POCUS-measured muscle thickness than with CT-derived SMI. To conclude, this study indicated a strong correlation between combined muscle thickness measurements performed by POCUS- and CT-derived SMI and measurements of muscle strength. These results suggest that handheld ultrasound is a valid tool for the assessment of skeletal muscle status.

## 1. Introduction

In recent decades, a growing body of evidence has recognized the importance of the role of skeletal muscle mass in physical health and recovery from diseases [1]. Currently, dual-energy X-ray absorptiometry (DXA), bioelectrical impedance analysis (BIA), and the golden-standard technique of computed tomography (CT) on the third lumbar level are often used to measure total skeletal muscle mass [2,3,4,5,6]. However, an easy-to-use, reliable, user-friendly tool to assess skeletal muscle mass is lacking in clinical practice.

Generalized loss of skeletal muscle mass and strength, also known as sarcopenia, is known to be associated with an increased risk of morbidity and mortality in patients with cancer [3,6,7,8,9]. Since sarcopenia covers both loss of muscle mass and loss of muscle strength, the easiest method to estimate a patient’s potential loss of muscle force is by measuring handgrip strength [10]. However, although handgrip strength is a noninvasive, inexpensive measurement, the relationship between muscle strength and muscle mass is not linear. Therefore, the European Working Group on Sarcopenia in Older People (EWGSOP) recommends using the presence of both low muscle mass and low muscle strength to diagnose sarcopenia [6,11].

However, the diagnostic tools to assess skeletal muscle mass face some limitations in their use in daily clinical practice. For example, although CT scans are often performed as standard of usual care in preoperative cancer staging, automatic segmentation of muscles at the third lumbar level is not yet widely available. DXA scans are time-consuming, require ionizing radiation, and cannot be used as a bedside diagnostic tool. Lastly, although BIA is an easy-to-use and low-cost method for the estimation of fat-free mass, the reliability of BIA measurements can be influenced by measurement errors related to the instrument itself or because of changes in hydration, soft-tissue edema, ascites, exercise status, or food intake [4,12].

Because it can be used as a handheld device without exposing the patient to extra iodizing radiation, point-of-care ultrasound (POCUS) can potentially be used as a tool to assess total skeletal muscle mass [13,14,15,16]. Although ultrasound has previously been proven to be a reliable and valid tool for the assessment of muscle size at a certain level of the extremities in older adults [13], the literature correlating measurements of skeletal muscle thickness using POCUS with whole-body mass equivalents measured using techniques such as abdominal CT scans is scarce [13,17].

The primary goal of this study was to validate skeletal muscle thickness measurements performed by POCUS against the CT-derived skeletal muscle index (SMI) at the third lumbar level. The secondary aim of this study was to evaluate the correlation between measurements of muscle thickness by POCUS and CT-derived SMI with muscle strength as a representation of muscle function.

## 2. Materials and Methods

This study was executed as a part of the Muscle Power Study, a prospective cohort study designed to identify the risk factors and clinical impact of surgery-related muscle loss after major abdominal cancer surgery [18]. Adult patients scheduled for major open surgery based on a (suspected) underlying malignant disease were assessed for inclusion eligibility during the period from May 2019 to June 2021. Patients with a recent (<3 months) preoperative CT scan available were included in this study. Because of hospital policy limiting the access of researchers to the patient wards due to a peak in COVID-19 cases, recruitment of new patients was temporarily halted between February 2020 and September 2020.

This study was approved by the Medical Ethics Committee of the University Medical Center Groningen (UMCG), and written consent from all participating patients was obtained prior to surgery. This study protocol was registered within the International Clinical Trails Registry Platform (201800445, NL7505, version 1.0, 7 February 2019). Patient data were processed and electronically stored in agreement with the Declaration of Helsinki’s ethical principles for medical research involving human subjects (2013). Demographic data, including age, sex, weight, height, body mass index (BMI), medical history, and American Society of Anesthesiologists score (ASA score), were prospectively collected from patients’ charts.

### 2.1. Muscle Thickness Measurements Using POCUS

POCUS was used to measure the muscle thickness of the bilateral musculi (mm.) biceps brachii, mm. rectus femoris, and mm. vastus intermedius the day prior to surgery according to a predefined and published protocol [18]. In brief, POCUS measurements were performed using a handheld ultrasound system (Philips FUS6882 Lumify L12-4) by three researchers (R.H., J.H., and W.P.), who were trained by a musculoskeletal radiologist (A.V.). The transducer was placed perpendicular to the long axis of the muscles while the patient was lying supine in bed with arm and leg muscles extended and relaxed. The location of the measurement of the m. biceps brachii was at two-thirds of the distance between the elbow fold and the tip of the acromion. Measurements of the m. rectus femoris and m. vastus intermedius were performed halfway between the spina iliaca anterior superior and the patella (Figure 1). POCUS muscle measurements were repeated three times bilaterally by the same observer. All muscles measured using POCUS (i.e., the average of m. biceps brachii, m. rectus femoris, and m. vastus intermedius) were analyzed as individual muscle groups and as combined muscle groups. The average of the six measurements (three times bilaterally) per muscle group was used for analysis. Total POCUS muscle thickness was defined as the total added value of the averages of bilateral m. biceps brachii, m. rectus femoris, and m. vastus intermedius. The POCUS muscle index was defined as the total POCUS muscle thickness divided by squared body height (cm/m^2^).

### 2.2. Muscle Mass Measurements CT

To assess muscle mass using CT scans, in-house developed AI-assisted software (SarcoMeas v0.65) was used to delineate the abdominal muscle wall and psoas muscles at the third lumbar level, where both transverse processes were best visible. If necessary, the contours were manually adjusted by an experienced radiologist (A.V.). Within these contours, the muscle area was defined as all voxels with a radiodensity ranging from −29 to 150 Hounsfield units (HU; Figure 1). Consequently, to correct for patient height, the skeletal muscle index (SMI) was calculated by dividing the total measured skeletal muscle area on the third lumbar level in square centimeters (cm^2^) by the squared patient height in meters (m) [19,20,21,22]. CT scans acquired up to a maximum of 3 months before the POCUS measurements were used.

### 2.3. Muscle Strength Measurements

Squeeze and force measurements were, like the POCUS measurements, performed 1 day prior to surgery. Isometric muscle force of grip strength for the hand, elbow flexion and extension, and knee flexion and extension were measured using a handheld dynamometer according to a predefined protocol [18]. Each measurement was performed three times on the left and right extremities with 20 s intervals between the measurements. The maximum measured value in kilograms (kg) was used for analysis.

### 2.4. Study Endpoints

The primary endpoint of this study was the correlation between POCUS measurements of skeletal muscle thickness of the m. biceps brachii, m. vastus intermedius, and m. rectus femoris with total abdominal SMI measured using CT. The secondary endpoint was the correlation between muscle strength measurements of squeeze force and elbow and knee flexion and extension, with POCUS measurements of skeletal muscle thickness and CT-derived measurements of SMI.

### 2.5. Statistical Analysis

Descriptive statistics were used to express patient characteristics, demographic data, and body composition measurements. Continuous variables are presented as the mean ± standard deviation (SD) or as the median with interquartile range (IQR) depending on the distribution. The distribution of continuous variables was visually tested using histograms.

The intraclass correlation coefficient (ICC) was calculated to assess the intra-rater reliability of these three repeated POCUS measurements per muscle group.

Pearson’s correlation coefficient was calculated to evaluate the relationship between muscle thickness measured using POCUS with CT-derived SMI. To interpret Pearson’s correlation coefficients, the following reference values were used: *r* < 0.3 indicates a negligible correlation, *r* 0.3–0.5 indicates a weak correlation, *r* 0.5–0.7 indicates a moderate correlation, *r* 0.7–0.9 indicates a strong correlation, and *r* > 0.9 indicates a very strong correlation. All statistical analyses were performed using the Statistical Package for the Social Sciences (SPSS 28.0, SPSS, Chicago, IL, USA). A *p*-value < 0.05 was considered to indicate statistical significance.

## 3. Results

A total of 187 patients were included in the Muscle Power Study in the period from May 2019 to June 2021. A total of 22 (12%) of these 187 patients did not have a recent (<3 months) abdominal CT scan suitable for muscle measurements. Therefore, a total of 165 (88%) patients were included in the final analysis.

An overview of characteristics for the included patients is presented in Table 1. The mean (SD) age of the included patients was 65 ± 12 years, and 55% of patients were male. The mean BMI was 26.23 ± 4.85 kg/m^2^, and severe preoperative weight loss (≥10%) within the 6 months prior to the POCUS muscle measurements was present in 32 (19.4%) patients.

Included patients were scheduled for the following major surgical procedures: liver segment resection (23.6%), colorectal resection (21.8%), pancreatic resection (37.6%), cytoreductive surgery with hyperthermic chemotherapy (14.5%), and other (2.4%).

### 3.1. Patient Body Composition Measurements

An overview of anthropometrics and measurements of muscle mass and muscle strength is provided in Table 2. Mean muscle thicknesses of the mm. biceps brachii, mm. vasti intermedius, and mm. recti femoris were 3.08 ± 0.60 cm, 1.33 ± 0.38 cm, and 1.32 ± 0.41 cm, respectively. The total mean thickness of bilateral mm. biceps brachii, mm. vasti intermedius, and mm. recti femoris (i.e., POCUS muscle thickness) was 11.49 ± 2.22 cm. The POCUS muscle index was 3.80 ± 0.66 cm/m^2^.

The mean time between the CT scans used for the analyses and POCUS measurements was 5 ± 3 weeks. The total CT-scan derived SMI was 43.75 ± 8.87 cm^2^/m^2^.

The mean maximum squeeze force of the included patients was 34 ± 12 kg. Elbow flexion and extension strengths were 21 ± 7 kg and 15 ± 5 kg, respectively. Knee flexion and extension strengths were 21 ± 6 kg and 25 ± 10 kg, respectively.

### 3.2. Correlation between Muscle Measurements

POCUS muscle thickness measurements indicated good intra-rater reliability, with intraclass correlations of 0.99 for m. biceps brachii, m. rectus femoris, and m. vastus intermedius measurements. Table 3 presents an overview of correlations between POCUS muscle measurements and SMI divided into total bilateral POCUS muscle thickness measurements and separate unilateral POCUS muscle thickness measurements.

All POCUS measurements of the muscle thickness of the m. biceps brachii, m. rectus femoris, and m. vastus intermedius had a significant correlation with CT-derived SMI (*p* < 0.001). Bilateral m. biceps brachii thickness showed a moderate correlation (*r* = 0.66), m. rectus femoris thickness showed a weak correlation (*r* = 0.48), and m. vastus intermedius showed a moderate correlation (*r* = 0.59). The strongest correlation between POCUS muscle measurements and CT-derived SMI was observed for the total POCUS muscle thickness of the total average of the mm. biceps brachii, mm. recti femoris, and mm. vasti intermedius (*r* = 0.73). A graphical overview of correlations between POCUS muscle measurements and SMI is presented in Figure 2.

Regarding muscle strength, significant correlations were observed between muscle force measurements of the upper and lower extremities and total POCUS muscle thickness and CT-derived SMI (*p* < 0.001; Table 4). Muscle strength had a stronger correlation with POCUS-measured muscle thickness than with CT-derived SMI (squeeze force: *r* = 0.61, *p* < 0.001 and *r* = 0.54, *p* < 0.001, respectively; Table 4).

## 4. Discussion

The results of this study revealed a strong correlation between combined arm and leg muscle thickness measurements performed using handheld POCUS and CT-derived SMI at the third lumbar level. This finding suggests that both measures are valid indicators of the total skeletal muscle mass of a patient. Muscle strength indicated a moderate correlation with POCUS measurements and a low to moderate correlation with CT measurements. These combined results suggest that handheld POCUS might be a valid and user-friendly tool to assess total skeletal muscle status in clinical practice.

Traditionally, assessment of skeletal muscle status is often performed via several devices such as DXA, BIA, and CT scans. However, limitations such as costs, ionizing radiation, and limited availability, and interfering factors such as edema or ascites make these devices less suitable for muscle mass assessment in clinical practice [4,12,13]. As such, in this study, we focused on the assessment of skeletal muscle mass by using POCUS as a bedside tool. A previously conducted systematic review demonstrated that POCUS measurements of the muscle thickness of the m. biceps brachii, m. rectus femoris, and m. vastus intermedius strongly correlate with CT scan measurements of these muscles on the same level [13]. However, since the SMI measured on the third lumbar level in CT scans is considered a derivate of generalized skeletal muscle status [5], the question arose as to whether these peripheral extremity measurements also correlate with generalized muscle status.

In this study, all individual measured muscle groups of the upper and lower extremities presented a moderate correlation with SMI. Although the majority of studies, including the present research, that used ultrasound for muscle measurements were performed on the quadriceps muscle [4,14,23,24,25], the best correlation between POCUS muscle measurements and overall skeletal muscle status was found in studies that combined muscle measurements of both the upper and the lower extremities [17,26]. For example, Paris et al. (*n* = 96) and Lambell et al. (*n* = 50) showed the strongest correlation between POCUS muscle measurements of the upper and lower extremities combined with fat-free mass measured by DXA or total skeletal muscle area on the third lumbar level measured by CT in critically ill patients [17,26]. 

Another method for assessment of skeletal muscle status is by measuring skeletal muscle functionality or muscle strength. However, as mentioned, muscle strength does not depend solely on muscle mass, and the EWGSOP, therefore, recommends using the presence of both low muscle mass and low muscle strength to diagnose sarcopenia [11]. In addition, a previous study determined that measurements of handgrip strength are less suitable for reflecting the changes in muscle mass and strength which occur as a result of physical exercise training programs or after surgical procedures [27]. Our study presented moderate correlations between muscle force mainly of the upper extremity with muscle thickness measured by POCUS and CT-derived SMI. Other studies conducted by Madden et al. (*n* = 150) and Strasser et al. (*n* = 52) found similar correlations between measurements of muscle mass performed using BIA and POCUS of the lower extremities with handgrip strength and maximum contraction force of the lower extremities [14,28]. However, to the best of the researchers’ knowledge, no other study has considered the correlation between muscle strength of the upper and lower extremities with both POCUS measurements and CT scan measurements. 

Since muscle force represents the functional capacity of a patient’s muscle mass, the moderate correlation between the total POCUS muscle thickness and various muscle strength parameters of both the upper and the lower extremities suggests that POCUS measurements provide a valuable tool for assessment of patients’ total muscle quantity and muscle quality in the outpatient clinic setting. In particular, in the context of frail elderly or in-hospital patients, it could be important to assess muscle functionality as part of personalized risk management [29,30,31]. Indeed, in recent years, more attention has been paid to the identification of “modifiable factors” as risk stratification for deviant surgical outcomes, and optimalization programs under the name of “prehabilitation” have been started [32]. At present, these prehabilitation programs often consist of identifying six factors: physical fitness, malnutrition, iron deficiency, anemia, frailty, intoxications, and psychological resilience [33,34]. By optimizing these risk factors, patients are assisted in enhancing their physical fitness before initiating highly demanding surgical treatments [33,34]. To date, the exact minimum time necessary to improve physical capacity is unknown. However, a study conducted by van Wijk et al. (*n* = 26) showed a significant increase of aerobic capacity after 4 weeks of training for high-risk patients scheduled for oncologic liver or pancreatic resection [35]. Additionally, Moug et al.’s (*n* = 44) randomized controlled trial found an improvement to the CT-derived psoas muscle index in 65% of patients who participated in a 13–17-week prehabilitation program while they received neoadjuvant chemoradiotherapy for rectal cancer, whereas 67% of the patients in the control group experienced a decrease in muscle mass [36]. Since the present study found a strong correlation between POCUS muscle measurements and CT muscle measurements and a moderate correlation between muscle strength measurements, the assessment of skeletal muscle mass by means of POCUS could be of additional value for individualizing these specific prehabilitation programs.

Nonetheless, this study had some limitations. Although a high intra-rater reliability was found with intraclass correlations of the POCUS measurements above 0.99, due to the study design, we were not able to test inter-rater reliability among the three researchers. However, previous studies have already identified high inter-rater reliability in POCUS measurements of skeletal muscles according to predefined protocols [13,17]. Thus, a strength of this study is that, despite all the POCUS measures being performed by three different researchers according to a predefined protocol, strong correlations with CT-derived SMI was nonetheless found; this suggests that POCUS skeletal muscle measurements are clinically applicable since they can be performed by multiple healthcare professionals.

To speculate, given that POCUS is an easy-to-use, handheld device that was utilized by three different researchers 1 day prior to surgery in the current study, implementation in the daily practices of general practitioners, physiotherapists, dieticians, and medical specialists could contribute to identification of individuals at risk for higher morbidity at a much earlier stage of treatment [3,6,7,8,9]. In the current study, total POCUS muscle thickness had the strongest correlation (*r* = 0.73) with CT-derived SMI. It is possible that, as a recently published study found a significant association between lower masseter muscle thickness as a derivate of low generalized muscle mass with deviant surgical outcomes after vascular surgery, adding POCUS masseter muscle measurements to the total POCUS muscle thickness may improve the correlation with CT-derived SMI [37]. On the other hand, given the limited time available in clinical practice, one could also consider measuring only m. biceps brachii thickness (*r* = 0.66) because measuring both the arm and the leg muscles is more time-consuming. However, more research is needed to determine the cutoff values for low muscle thickness and to study the association between muscle thickness and frailty or sarcopenia.

## 5. Conclusions

To conclude, this study found a strong correlation between POCUS measurements of the mm. biceps brachii, mm. recti femoris, and mm. vasti intermedius and SMI derived from abdominal CT scans. A correlation was identified between POCUS muscle thickness measurements and muscle strength. These results suggest that handheld ultrasound is a clinically applicable and valid tool for assessment of skeletal muscle status in both ambulatory and in-hospital clinical settings.

## Figures and Tables

**Figure 1 jcm-11-06606-f001:**
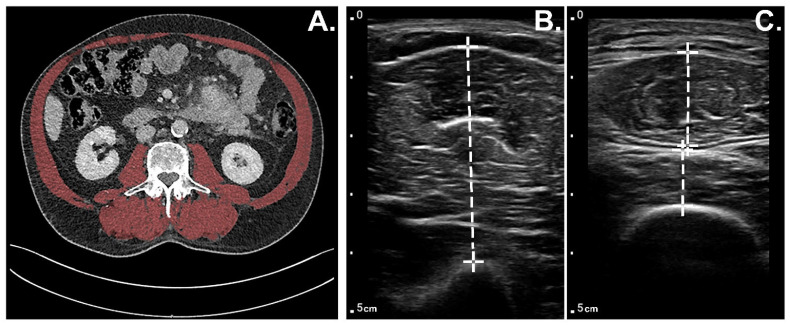
Example muscle mass measurements. (**A**) Total skeletal muscle area on the third lumbar level on a CT scan is marked in red. (**B**) POCUS measurement (in centimeter) of the thickness of the m. biceps brachii. (**C**) POCUS measurements of the thickness of the m. rectus femoris (upper) and m. vastus intermedius (lower).

**Figure 2 jcm-11-06606-f002:**
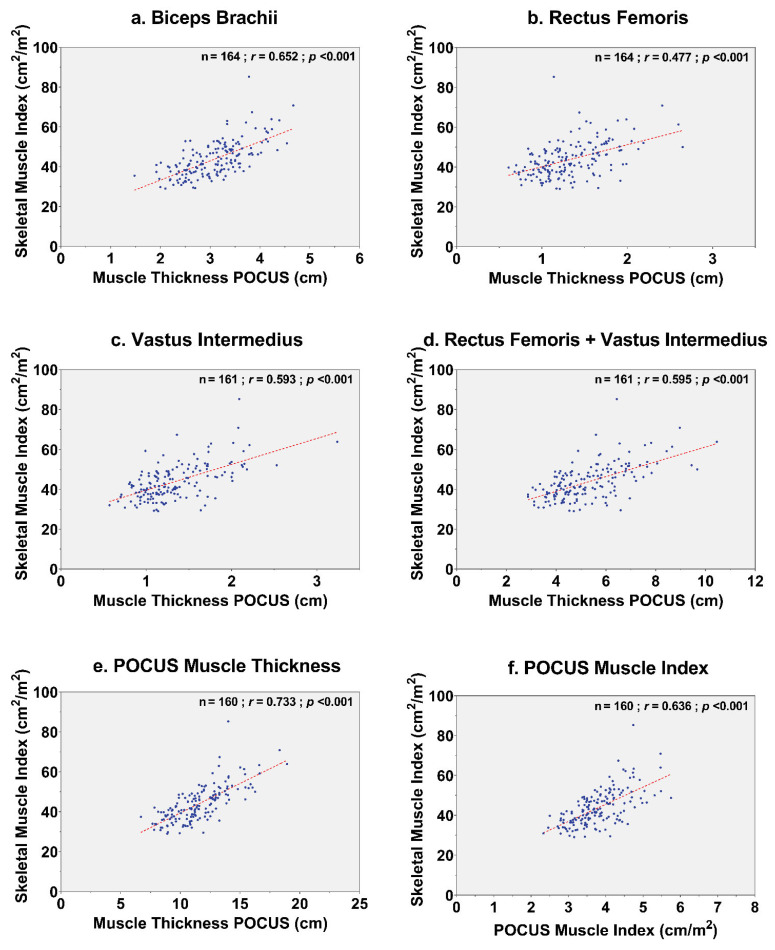
Correlation plots of POCUS muscle measurements with skeletal muscle index. Correlation between POCUS muscle measurements of the (**a**) biceps brachii, (**b**) rectus femoris, (**c**) vastus intermedius, (**d**) rectus femoris and vastus intermedius, (**e**) total POCUS muscle thickness, and (**f**) POCUS muscle index with CT-derived SMI. Abbreviations: POCUS = point-of-care ultrasound; POCUS muscle thickness = the added value of the averages of bilateral m. biceps brachii, m. rectus femoris, and m. vastus intermedius; POCUS muscle index = POCUS muscle thickness divided by squared body height.

**Table 1 jcm-11-06606-t001:** Patient characteristics.

Patient Characteristics	Total (*n* = 165)
Age, years	65 ± 12
Sex, male	91 (55.2%)
ASA grade ≥3	41 (24.8%)
PG-SGA SF ≥4	54 (32.7%)
Preoperative weight loss	
<5%	107 (64.8%)
5–10%	26 (15.8%)
≥10%	32 (19.4%)
Comorbidities	
Any comorbidity	103 (62.4%)
Cardiac	44 (26.7%)
Diabetes mellitus	22 (13.3%)
Hypertension	47 (28.5%)
Pulmonal	28 (17%)
Renal	7 (4.2%)
Type of tumor	
Appendix	4 (2.4%)
Bile ducts	31 (18.8%)
Colon	18 (10.9%)
Colorectal liver metastases	14 (8.5%)
Colorectal peritoneal metastases	4 (2.4%)
Liver	8 (4.8%)
Pancreas	42 (25.5%)
Pseudomyxoma peritonei	5 (3%)
Rectum	22 (13.3%)
Small bowel	5 (3.0%)
Other	12 (7.3%)
Prior oncologic treatment	53 (32.1%)
Neoadjuvant chemotherapy	46 (27.9%)
Neoadjuvant radiotherapy	31 (18.8%)
Type of scheduled operation	
Liver segment resection	39 (23.6%)
Colorectal resection	36 (21.8%)
Pancreatic resection	62 (37.6%)
CRS with HIPEC	24 (14.5%)
Other	4 (2.4%)

Postoperative outcomes	
Length of hospital stay, days	12 (8–19)
Occurrence of any complication	116 (70.3%)
Occurrence of severe complication (Clavien–Dindo ≥3)	37 (22.4%)
Readmission <30 days	26 (15.8%)
Mortality <30 days	3 (1.8%)

Data are presented as the mean ± SD, median (IQR), or number (%). ASA = American Society of Anesthesiologists score, PG-SGA SF = Patient-Generated Subjective Global Assessment Short Form; CRS with HIPEC = cytoreductive surgery with heated intraperitoneal chemotherapy.

**Table 2 jcm-11-06606-t002:** Body composition measurements.

Body Composition Measurements	Total (*n* = 165)
Anthropometrics	
Height, m	1.74 ± 0.10
Weight, kg	79.56 ± 17.04
Body mass index, kg/m^2^	26.23 ± 4.85
Computed tomography	
Time between CT and POCUS, weeks	5 ± 3
Muscle area	
Total skeletal muscle index, cm^2^/m^2^	43.75 ± 8.87
Abdominal muscle wall index, cm^2^/m^2^	38.17 ± 7.64
Psoas muscle index, cm^2^/m^2^	5.58 ± 1.71
Point-of-care ultrasound	
Muscle thickness	
Biceps brachii, cm	3.08 ± 0.60
Recuts femoris, cm	1.33 ± 0.38
Vastus intermedius, cm	1.32 ± 0.41
Rectus femoris + vastus intermedius, cm	5.29 ± 1.43
POCUS muscle thickness, cm	11.49 ± 2.22
POCUS muscle index, cm/m^2^	3.80 ± 0.66
Muscle strength	
Squeeze force, kg	34 ± 12
Elbow flexion, kg	21 ± 7
Elbow extension, kg	15 ± 5
Knee flexion, kg	21 ± 6
Knee extension, kg	25 ± 10

Data are presented as the mean ± SD, median (IQR), or number (%). Abbreviations: POCUS = point-of-care ultrasound; POCUS muscle thickness = the added value of the averages of bilateral m. biceps brachii, m. rectus femoris, and m. vastus intermedius; POCUS muscle index = POCUS muscle thickness divided by squared body height.

**Table 3 jcm-11-06606-t003:** Correlation between preoperative muscle thickness measured by POCUS and CT-derived Skeletal Muscle Index.

POCUS Measurements (cm)	N	CT-Derived Skeletal Muscle Index (cm^2^/m^2^)	
		Pearson’s Correlation *r*	*p*-Value

Bilateral			
Biceps brachii	164	0.66	<0.001
Rectus femoris	164	0.48	<0.001
Vastus intermedius	161	0.59	<0.001
Rectus femoris + vastus intermedius	161	0.60	<0.001
POCUS muscle thickness, cm	160	0.73	<0.001
POCUS muscle index, cm/m^2^	160	0.64	<0.001
Dominant side			
Biceps brachii	165	0.64	<0.001
Rectus femoris	164	0.45	<0.001
Vastus intermedius	161	0.57	<0.001
Rectus femoris + vastus intermedius	161	0.58	<0.001
POCUS muscle thickness, cm	161	0.72	<0.001
POCUS muscle index, cm/m^2^	161	0.62	<0.001
Nondominant side			
Biceps brachii	164	0.64	<0.001
Rectus femoris	164	0.47	<0.001
Vastus intermedius	161	0.57	<0.001
Rectus femoris + vastus intermedius	161	0.58	<0.001
POCUS muscle thickness, cm	160	0.72	<0.001
POCUS muscle index, cm/m^2^	160	0.62	<0.001

Correlation between muscle mass measurements performed by POCUS with CT-derived skeletal muscle index measured on the third lumbar level. Abbreviations: POCUS = point-of-care ultrasound; POCUS muscle thickness = the added value of the averages of bilateral m. biceps brachii, m. rectus femoris, and m. vastus intermedius; POCUS muscle index = POCUS muscle thickness divided by squared body height.

**Table 4 jcm-11-06606-t004:** Correlation between muscle strength measurements and total POCUS muscle thickness and CT-derived skeletal muscle index measured on the third lumbar level.

	N	Total POCUS Muscle Thickness (cm)			CT-Derived Skeletal Muscle Index (cm^2^/m^2^)		
		Pearson’s Correlation *r*	95% CI	*p*-Value	Pearson’s Correlation *r*	95% CI	*p*-Value
Squeeze force	158	0.61	0.50–0.70	<0.001	0.54	0.42–0.64	<0.001
Elbow flexion	157	0.64	0.54–0.73	<0.001	0.58	0.46–0.67	<0.001
Elbow extension	157	0.65	0.53–0.72	<0.001	0.57	0.47–0.68	<0.001
Knee flexion	158	0.51	0.39–0.62	<0.001	0.41	0.27–0.54	<0.001
Knee extension	158	0.33	0.18–0.46	<0.001	0.29	0.14–0.43	<0.001

POCUS = point-of-care ultrasound; POCUS muscle thickness = the added value of the averages of bilateral m. biceps brachii, m. rectus femoris, and m. vastus intermedius; POCUS muscle index = POCUS muscle thickness divided by squared body height.

## Data Availability

The datasets generated or analyzed during the current study are not publicly available as the data are linked to a vulnerable patient population, but these are available from the corresponding author on reasonable request.

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
