# Peer review of "Thickness of Biceps and Quadriceps Femoris Muscle Measured Using Point-of-Care Ultrasound as a Representation of Total Skeletal Muscle Mass"

_jcm, 2022, doi:10.3390/jcm11226606_

Round 1

Reviewer 1 Report

The authors examined the validity of skeletal muscle thickness measurements by POCUS against the CT-derived skeletal muscle index (SMI). Results indicated the clinical availability of POCUS measurement as easier detection of muscle status against the SMI derived from CT. Therefore, it will be favorable and benefit a wide range of clinical use.   

1.      One point, I feel something wrong to use the word “sarcopenia”. Basically, sarcopenia is an ageing relate decrease of muscle mass and function with closely related frailty. Thus, it should be distinguished from the disease, injury and surgery-related muscle loss. The material of this study is a surgery-derived muscle loss after major abdominal cancer surgery. Therefore, the first two sentences in the abstract are inappropriate and not necessary.

2.      In the above regard, the context of the first two paragraphs of the Introduction should be better to adjust to the primary endpoint of this study (2. 4). Because the results of this study is not directly related to the sarcopenia, although this POCUS measurement may be available for the evaluation of sarcopenia. At first, it should uphold the surgery-related loss of muscle mass and secondary sarcopenia is easy to accept.   

Figure 1 need more information. What is a red color in panel A. It maybe includes erector muscles of spine. Vertical scale unit of B and C is necessary. Is direct comparison between B and C possible?

Author Response

Reviewer #1

The authors examined the validity of skeletal muscle thickness measurements by POCUS against the CT-derived skeletal muscle index (SMI). Results indicated the clinical availability of POCUS measurement as easier detection of muscle status against the SMI derived from CT. Therefore, it will be favourable and benefit a wide range of clinical use.   

  1. One point, I feel something wrong to use the word “sarcopenia”. Basically, sarcopenia is an ageing relate decrease of muscle mass and function with closely related frailty. Thus, it should be distinguished from the disease, injury and surgery-related muscle loss. The material of this study is a surgery-derived muscle loss after major abdominal cancer surgery. Therefore, the first two sentences in the abstract are inappropriate and not necessary.
  2. In the above regard, the context of the first two paragraphs of the Introduction should be better to adjust to the primary endpoint of this study (2. 4). Because the results of this study is not directly related to the sarcopenia, although this POCUS measurement may be available for the evaluation of sarcopenia. At first, it should uphold the surgery-related loss of muscle mass and secondary sarcopenia is easy to accept.   

Authors’ reply: We thank the reviewer for his/her feedback and agree on the first two items raised by the reviewer which states that the paper focusses mainly on generalized muscle status and not on sarcopenia itself.

We think that for providing enough background regarding the importance of assessing muscle status, it is of added value to include a passage on sarcopenia in the introduction of the manuscript. Therefore, as suggested by the reviewer, we replaced the term ‘sarcopenia’ in the abstract (L17-18). And we rearranged the introduction, so at first paragraphs are more adjusted tot the primary endpoint of the study (to validate POCUS as an measure to assess muscle status in when compared with CT-derived skeletal muscle index).

  1. Figure 1 need more information. What is a red color in panel A. It maybe includes erector muscles of spine. Vertical scale unit of B and C is necessary. Is direct comparison between B and C possible?

Authors’ reply: As suggested, we’ve added an explanation for the red marked area on Figure 1A. As the reviewer rightfully points out: the paravertebral muscles were also marked in red because this can also be considered as ‘the posterior abdominal wall’. Therefore, these muscles are also included for the assessment of total skeletal muscle status on the ‘abdominal level’ of L3. Nonetheless, to avoid confusion we’ve corrected the terminology of ‘total abdominal muscle mass’ to total skeletal muscle area on the third lumbar level.

Reviewer 2 Report

Good clinical applicable study. Some comments:

Line 34: Key wards: ultrasound to be deleted because POCUS carries the same meaning.

Line 75: MUSCLE POWER STUDY. Why it is written in capital???

Can you add some economic aspects as the cost of CT vs the US in your country in Euro?

Is the POCUS carried out by a sonographer or by a radiologist in your place?

Is there any follow-up for those patients after operations regarding morbidity and/or mortality? This will be of great value if you cam add.

Author Response

Reviewer #2

Good clinical applicable study. Some comments:

Line 34: Key wards: ultrasound to be deleted because POCUS carries the same meaning.

Authors’ reply: We thank the reviewer for his/her suggestion and removed the word ‘ultrasound’ from the list of keywords (L34).

Line 75: MUSCLE POWER STUDY. Why it is written in capital???

The words MUSCLE POWER STUDY were written in capital because this is the original name in which the study was carried out. Also the study protocol was published previously under the name MUSCLE POWER STUDY (Hentzen, J.E.K.R.; van Wijk, L.; Buis, C.I., et al. Impact and Risk Factors for Clinically Relevant Surgery-Related Muscle Loss in Patients after Major Abdominal Cancer Surgery: Study Protocol for a Prospective Observational Cohort Study (MUSCLE POWER). Int. J. Clin. Trials 2019, 6, 138–146, doi:doi:http://dx.doi.org/10.18203/2349-3259.ijct20193217).

However, to avoid confusion for the readers we adjusted the capitals for ‘Muscle Power Study’ (L79, L165).

Can you add some economic aspects as the cost of CT vs the US in your country in Euro?

Authors’ reply: The costs to execute an abdominal CT scan depends very much on the country, indication for the scan, clinical setting in which the CT-scan is carried out (private or public clinic) and the assessment of the radiologist. It is difficult to provide exact numbers on differences in costs for CT versus ultrasound, however one can imagine that performing POCUS is an easier tool to use for bed-side assessment of skeletal muscle status. Thereby, when considering assessment of skeletal muscle mass, even though the CT scans are often performed as standard of usual care in preoperative cancer staging, automatic segmentation of muscles at the third lumbar level is not yet widely available. While our manuscript suggests that ultrasound as a modality to assess skeletal muscle status could be a valid tool in clinical practice.

Is the POCUS carried out by a sonographer or by a radiologist in your place?

Authors’ reply: For this study, the POCUS was carried out by three researchers with a Medical Degree (RH, JH, WP) who were trained by a musculoskeletal radiologist (AV) (L102,130). So for future research, or in clinical practice, the measurements of muscle thickness can eighter be carried out by radiologists, sonographers or even by physicians who are trained to perform the measurements.

Is there any follow-up for those patients after operations regarding morbidity and/or mortality? This will be of great value if you can add.

Authors’ reply: We thank the reviewer for his/her suggestion regarding the added value of postoperative surgical outcomes regarding morbidity and/or mortality. Unfortunately, the design of the study, by including patients with a high variety in oncological diagnoses, does not allow us to perform correlation analyses on the association between low skeletal muscle status measured by ultrasound and the occurrence of deviant surgical outcomes. However, to provide a complete description of our patient population, we have added descriptives on postoperative surgical outcomes to Table 1.